# Nutraceuticals/Drugs Promoting Mitophagy and Mitochondrial Biogenesis May Combat the Mitochondrial Dysfunction Driving Progression of Dry Age-Related Macular Degeneration

**DOI:** 10.3390/nu14091985

**Published:** 2022-05-09

**Authors:** Lidianys María Lewis Luján, Mark F. McCarty, James J. Di Nicolantonio, Juan Carlos Gálvez Ruiz, Ema Carina Rosas-Burgos, Maribel Plascencia-Jatomea, Simon Bernard Iloki Assanga

**Affiliations:** 1Department of Research and Postgraduate in Food, University of Sonora, Blvd. Luis Encinas y Rosales S/N, Col. Centro, Hermosillo 83000, Mexico; lidianys1@yahoo.es (L.M.L.L.); carina.rosas@unison.mx (E.C.R.-B.); maribel.plascencia@unison.mx (M.P.-J.); 2Catalytic Longevity Foundation, San Diego, CA 92109, USA; markfmccarty@gmail.com; 3St. Luke’s Mid America Heart Institute, Kansas City, MO 64111, USA; jjdinicol@gmail.com; 4Department of Biological Chemical Sciences, Sonora University, Blvd. Luis Encinas y Rosales, Col. Centro, Hermosillo 83000, Mexico; juan.galvez@unison.mx

**Keywords:** nutraceuticals, age-related macular degeneration, mitochondrial biogenesis, Sirt1, AMPK, Nrf2, ferulic acid, melatonin, berberine, astaxanthin

## Abstract

In patients with age-related macular degeneration (AMD), the crucial retinal pigment epithelial (RPE) cells are characterized by mitochondria that are structurally and functionally defective. Moreover, deficient expression of the mRNA-editing enzyme Dicer is noted specifically in these cells. This Dicer deficit up-regulates expression of Alu RNA, which in turn damages mitochondria—inducing the loss of membrane potential, boosting oxidant generation, and causing mitochondrial DNA to translocate to the cytoplasmic region. The cytoplasmic mtDNA, in conjunction with induced oxidative stress, triggers a non-canonical pathway of NLRP3 inflammasome activation, leading to the production of interleukin-18 that acts in an autocrine manner to induce apoptotic death of RPE cells, thereby driving progression of dry AMD. It is proposed that measures which jointly up-regulate mitophagy and mitochondrial biogenesis (MB), by replacing damaged mitochondria with “healthy” new ones, may lessen the adverse impact of Alu RNA on RPE cells, enabling the prevention or control of dry AMD. An analysis of the molecular biology underlying mitophagy/MB and inflammasome activation suggests that nutraceuticals or drugs that can activate Sirt1, AMPK, Nrf2, and PPARα may be useful in this regard. These include ferulic acid, melatonin urolithin A and glucosamine (Sirt1), metformin and berberine (AMPK), lipoic acid and broccoli sprout extract (Nrf2), and fibrate drugs and astaxanthin (PPARα). Hence, nutraceutical regimens providing physiologically meaningful doses of several or all of the: ferulic acid, melatonin, glucosamine, berberine, lipoic acid, and astaxanthin, may have potential for control of dry AMD.

## 1. The Complex Molecular Biology Underlying the Pathogenesis of Dry AMD

Retinal pigment epithelium, which separates the neural retina from the underlying blood vessel-rich choroid, performs a number of tasks required for healthful ocular functions: phagocytizing and degrading the distal ends of photoreceptors while regenerating 11-cis retinal; regulating the flux of molecules from the choroid to the retina; and producing trophic factors which sustain the survival of retinal neurons while preventing over-proliferation of choroid blood vessels. The progressive loss of RPE cells over time in aging humans is responsible for the most common cause of irreversible visual impairment, so-called dry age-related macular degeneration (AMD)—also known as geographic atrophy.

Recent evidence points to the formation of NLRP3-dependent inflammasomes in retinal pigment epithelium (RPE) as a key factor driving RPE cell death in dry AMD. In particular, a series of elegant investigations, involving study of retinal pigment epithelium from deceased patients with dry AMD, studies with human and mouse RPE-derived cell lines, and studies in mice, have characterized a complex mechanism, consistent with the results of these studies, by which such inflammasomes can arise and induce the death of RPE cells [1,2,3,4,5,6]. For reasons yet to be clarified, expression of the microRNA-processing enzyme DICER1 is notably depressed in the RPE of patients with GA [1]. Moreover, knockdown of DICER1 leads to death of RPE in mice. While DICER1 plays a key role in generation of micro-RNAs, knockdown of other enzymes required for microRNA generation fails to induce death in RPE cells. However, DICER1 also functions to cleave the Alu RNA retrotransposon, and, as a consequence of DICER1 deficiency, this accumulates to excessive levels in the RPE [7]. In RPE cell lines, Alu RNA excess has been shown to lead to cell death which is dependent on the activation of NLRP3 inflammasomes. These generate interleukin-18, which acts in an autocrine fashion via MyD88 to induce cell death via a mechanism dependent on Fas ligand and caspase-8 [4]. These mechanisms seem likely to be pertinent to clinical AMD, as RPE from dry AMD patients displays increased levels of NRLP3 inflammasomes, IL-18, and caspases 1 and 8 [4].

The excess of Alu RNA induces structural and functional damage to RPE mitochondria that plays an obligate role in inflammasome activation. This damage entails the opening of mitochondrial permeability transition pores associated with increased mitochondrial generation of oxidants, a reduction in mitochondrial membrane potential, and escape of mitochondrial DNA (mtDNA) into the cytoplasm. Increased mitochondrial oxidant generation provides the oxidant stress required for NLRP3 inflammasome activation, as mitochondrially -targeted antioxidants—but not inhibitors of NADPH oxidase—block the impact of Alu RNA excess on inflammasome activation and cell death in RPE cells [2]. However, cytoplasmic mtDNA also plays a role in Alu RNA-mediated inflammasome activation. However, cytoplasmic mtDNA interacts with and activates cyclic GMP-AMP synthase (cGAS), which via interferon response factor 3 (IRF3) activation, promotes transcription of interferon-β [6]. Autocrine signaling by the latter drives non-canonical activation of NLRP3 inflammasomes via a mechanism dependent on caspase 4 expression/activation and on the presence of gasdermin [2].

Alu RNA also promotes inflammasome priming via NF-kappaB activation, another possible consequence of cGAS activation [3,8]. Activation of NF-kappaB might also reflect interaction of Alu RNA with double-stranded RNA receptors [9]. Stimulation of P2X7 receptors—leading to an efflux of cellular potassium also participates in Alu RNA-induced inflammasome activation. Extracellular ATP is the canonical mediator of P2X7 activation; however, whether ATP efflux mediates Alu RNA-induced P2X7 activation remains unclear. ATP release through gasdermin-mediated pore formation does not appear to play a role in this regard, even though the presence of gasdermin is required [6]. It has been suggested that Alu RNA may mediate P2X7 activation via an intracellular effect on the receptor, rather than ATP release [10]. Finally, activated NLRP3 generates interleukin-18 (IL-18), which via autocrine signaling leads to caspase-8-mediated apoptosis of RPE cells [4]. Figure 1 offers a simplified depiction of how DICER1 deficit and consequent Alu RNA excess lead to mitochondrial damage, inflammasome activation, and IL-18-mediated RPE cell death.

It is reasonable to presume that certain environmental factors linked to increased risk for GA, do so by amplifying some of the signaling mechanism outlined here. In particular, amyloid beta, complement component C5a, and the bisretinoid A2E have been found to promote NLRP3 inflammasome formation in ARPE-19 cells [11,12,13].

Accumulating evidence indicates that mitochondria are indeed structurally and functionally impaired in the RPE of AMD patients [14,15,16,17]. A loss of mitochondrial mass disrupted internal structure, diminished expression of electron transport chain proteins, and elevated oxidation of mtDNA have all been reported; the extent to which Alu RNA excess drives this phenomenon remains unclear [14,16]. These findings have prompted a number of investigators to suggest that measures which improve mitochondrial structure and function might prove useful for preventing and controlling dry AMD [14,17,18]. It is reasonable to suspect that the ability of Alu RNA to induce mitochondrial oxidant stress and promote extrusion of mtDNA is not just an acute effect observable with healthy mitochondria but may be a gradually evolving phenomenon in which mitochondrial function is progressively disrupted. If so, then insuring that RPE mitochondria are structurally and functionally sound by promoting mitophagy of unsound mitochondria, coupled with biogenesis of new mitochondria, might ameliorate the contribution of disrupted mitochondria to NLRP3 inflammasome activation and the consequent apoptotic death of RPE cells. And, measures which work in complementary ways to suppress NLRP3 inflammasome activation might also contribute to a rational strategy for combatting dry AMD. As explained below, nutraceuticals or drugs which boost the activity of Sirt1, AMPK, Nrf2, and PPARalpha could be expected to achieve these goals.

## 2. Regulation of Mitophagy, Mitochondrial Biogenesis, and Inflammasome Activation

The type III deacetylase Sirt1 and AMPK-activated kinase (AMPK) collaborate in the activation of both autophagy and mitophagy, while promoting mitochondrial biogenesis by boosting the activity of PPARg coactivator-1alpha (PGC-1a). They also can boost each other’s activity. AMPK increases expression of nicotinamide ribosyltransferase (NAMPT), which promotes synthesis of Sirt1’s obligate substrate NAD+. Sirt1, in turn, deacetylates and thereby stabilizes LKB1, which can phosphorylate and thereby activate AMPK [19,20,21,22,23]. With respect to autophagy, Sirt1 boosts the activity of number of proteins which participate in autophagosome formation; it also promotes the activity of Rab7, a G protein that enables fusion of autophagosomes and lysosomes [24,25,26]. Via deacetylation of the transcription factor FOXO1, Sirt1 enhances transcription of a number of genes which code for mediators of autophagy [26]. AMPK promotes autophagy by conferring an activating phosphorylation on ULK1, while inhibiting and suppressing the anti-autophagic activity of mTORC1 [27,28,29]. In regard to mitophagy, Sirt1 and AMPK collaborate in enhancing the expression of Parkin and Pink1, proteins which detect defective mitochondria with diminished membrane potential and mark them for incorporation into autophagic vacuoles; the mechanism of this effect is described below.

The coactivator activity of PGC-1a works in multiple ways to stimulate the synthesis of proteins that participate in mitochondrial biogenesis (MB), through its interactions with the transcription factors NRF-1, estrogen-related receptor-alpha (ERRα), and PPARα [30]. Aided by PGC-1α, NRF-1 promotes the transcription of genes coding for enzymes that mediate replication of mtDNA, such as Tfam, NRF-2, and TFB1 [31]. PGC-1α’s interaction with ERRα boosts expression of Sirt3; the latter alters the transcriptional activity of FOXO3a, enabling it to boost transcription of mtDNA genes coding for certain proteins in the mitochondrial electron transport chain, as well as for the mitochondrial form of superoxide dismutase (SOD2) [32,33,34,35]. Sirt3 also enhances the activity of SOD2 by deacetylating it [36,37]. Hence, Sirt3 exerts a profound antioxidant effect within mitochondria. Importantly, Sirt3-deacetylated FOXO3a also promotes transcription of genes coding for Parkin and Pink1, the mediators of mitophagy [38,39,40]. Finally, via interaction with the transcription factor PPARα, PGC-1α boosts expression of the uncoupling protein UCP-2 (whose activity decreases mitochondrial superoxide generation) and a number of enzymes, including carnitine palmitoyltransferase-1, required for mitochondrial oxidation of fatty acids [41,42,43].

The contribution of Sir1 and AMPK to MB hinges on their ability to enhance both the expression and the activity of PGC-1α. Phosphorylation of PGC-1α by AMPK is a pre-requisite for its deacetylation by Sirt1; these effects importantly enhance PGC-1α’s activity [44]. Sirt1 also promotes PGC-1α’s expression at the transcriptional level by reducing activity of NF-kB, which can bind to the promoter of the PGC-1α gene and inhibit its transcription [45,46].

The transcriptional factor Nrf2 participates in MB by driving transcription of the transcription factor NRF-1, which, as we have seen, works with PGC-1a to promote expression of proteins vital for MB [47,48,49]. Sirt1 can enhance Nrf2’s activity in this regard via deacetylation [50]. The reader is referred to Figure 2 for a depiction of the mechanisms whereby Sirt1, AMPK, Nrf2, and PPARα interact in the promotion of MB, while also enhancing expression of the mediators of mitophagy, Parkin and Pink1.

It is reasonable to suspect that activation of autophagy/mitophagy and MB will lessen activation of NLRP3 inflammasomes in RPE cells by suppressing mitochondrial oxidant generation and the cytoplasmic release of mtDNA. However, Nrf2 and AMPK can also inhibit inflammasome activation via their effects on expression and activity of thioredoxin (TRX) and thioredoxin interacting proteins (TXNIPs). The formation of NLRP3 inflammasomes hinges on the interaction between TXNIP and NLRP3; in cells with good redox control, this interaction is suppressed by complex formation between TXNIP and TRX [51,52]. In the context of oxidative stress, reversible oxidation of TRX prevents its interaction with TXNIP, freeing the latter to participate in inflammasome formation. Nrf2 activity, promoted by Sirt1, induces expression of both TRX and the enzyme which restores its reduced form, thioredoxin reductase, thereby decreasing the free pool of TXNIP [53,54]. The contribution of AMPK to suppression of inflammasome formation hinges on its ability to decrease the expression of TXNIP at the transcriptional level, possibly reflecting its inhibitory effect on activity of the transcription factor carbohydrate response element-binding protein (ChREBP) [55,56]. AMPK also has the potential to accelerate the proteasomal degradation of TXNIP [57].

The foregoing explains how Sir1, AMPK, Nrf2 and PPARα can interact to replace damaged mitochondria with healthy new ones, while amplifying mechanisms which protect mitochondria from oxidative stress and diminish their production of oxidants. Additionally, via the modulation of TRX/TXNIP, they work in an additional way to blunt NLRP3 inflammasome formation. Importantly, there are nutraceuticals and drugs capable of boosting the activity of each of these key enzymes.

## 3. Nutraceuticals and Drugs Can Activate Sirt1, AMPK, Nrf2 and PPARα

While resveratrol has gained a considerable reputation as an agent that can activate Sirt1, poor pharmacokinetics have to this point prevented it from being clinically useful for this purpose [58]. Moreover, its ability to directly activate Sirt1 has been challenged [59]. A more practical choice in this regard is ferulic acid. Ferulic acid is produced when gut bacteria metabolize dietary anthocyanins; the latter are not absorbed intact and ferulic acid seems likely to mediate most if not all of their protective properties [60]. Ferulic acid also occurs, mostly in conjugated forms, in a wide range of whole grains, fruits, vegetables and nuts. In animal and cell-culture studies, ferulic acid has demonstrated anti-inflammatory and antioxidant effects in a wide range of disease models; in Chinese medicine, sodium ferulate has been widely used in cardiovascular medicine [60,61]. The piperazine salt of ferulate is used in China for treatment of diabetic nephropathy [62]. In a recent controlled clinical trial, the administration of 1000 mg ferulic acid daily to hyperlipidemics was found to improve serum lipid profiles, decrease systemic markers of oxidative stress, and lower plasma C-reactive protein by one-third [63].

Only recently has it been recognized that its utilities in these regards may reflect its ability to up-regulate Sirt1 expression quite markedly at both the mRNA and protein level; this effect appears to be broad in scope, as it has been reported so far in chondrocytes, skeletal muscle fibers, testes, neural stem cells and hepatocytes [64,65,66,67,68,69]. How it achieves this effect still remains mysterious. In light of its efficient absorption, and its demonstrated benefits in cardiovascular medicine, ferulic acid, or salts thereof, may have considerable clinical potential for boosting Sirt1 activity. Of possible pertinence to AMD is the fact that rich dietary sources of anthocyanins—notably bilberry extract—have been traditionally used for retinal protection [70,71,72]. Whether they are genuinely beneficial for the prevention of dry AMD has not yet received adequate study.

Also useful in for Sirt1 activation is melatonin, which increases transcription of the gene encoding for Sirt1 via the activation of the BMAL1 transcription factor [73,74]. Melatonin also increases Nrf2 expression by the same mechanism, and hence may have considerable potential for dry AMD control [75,76]. Melatonin administration has been found to protect the retina in a mouse model of dry AMD (superior cervical ganglionectomy) and in a rat model of accelerated senescence associated with AMD [77,78]. Favorable anecdotal clinical experience with supplemental melatonin in AMD has also been reported, but formal clinical trials evaluating its efficacy in this regard are lacking [79].

Sirt1 is susceptible to O-GlcNAcylation at Ser 549, and this post-translational modification has been found to amplify its deacetylase activity [80]. Oral or parenteral administration of glucosamine can increase the cellular pool of UDP-N-acetylglucosamine; this therefore up-regulates the O-GlcNAcylation of proteins, likely explaining the anti-inflammatory properties of supplemental glucosamine [81]. The ability of glucosamine to boost Sirt1 activity may hence explain recent epidemiology pointing to a lower risk for dry AMD in people who use this nutraceutical regularly; adjusted hazard rate for those using glucosamine for 3 or more years was found to be 0.493 (*p* = 0.003) [82]. Increased Sirt1 activity may also explain an increase in skeletal muscle MB and exercise performance in mice given oral glucosamine as an adjuvant to aerobic training [83].

Very recently, the bacterial ellagitannin metabolite urolithin A (which may mediate the protective benefits of pomegranate juice) has been shown to have potential for Sirt1 activation, as has the natural nicotinamide metabolite N1-methylnicotinamide, both of which unsurprisingly have anti-inflammatory properties [84,85,86,87,88]. Both urolithin A and MNA have recently become available in nutraceutical form, but have not yet been studied in regard to possible impact on AMD.

It is well know that the favorable impact of the drug metformin on diabetes control reflects its ability to activate AMPK [89,90]. Consistent with the thesis presented here, diabetics who use metformin have been reported to be at lower risk for AMD than diabetics who do not use this drug– albeit findings in this regard are not completely consistent [91,92,93,94]. The nutraceutical berberine, a key component of certain Chinese medicinal herbs, can also activate AMPK in a manner comparable to metformin, and is used widely in China for diabetes control [95,96,97,98,99].

A number of phytochemicals (known as phase 2 inducers) are capable of promoting the migration of Nrf2 to the nucleus—thereby promoting its transcriptional activity—by disrupting its binding to the cytoplasmic protein Keap1 [100,101]. The most clinically useful agents in this regard appear to be lipoic acid and sulforaphane—the latter can be supplied for clinical use in broccoli sprout extracts [102,103,104]. The ability of Sirt1 to deacetylate Nrf2 is complementary to the activity of phase 2 inducers in promoting the transcriptional activity of Nrf2 [50,105].

Drugs which can act as agonists for PPARα are used for the management of hyperlipidemia associated with metabolic syndrome; these include the fibrate drugs fenofibrate and gemfibrozil [106]. Curiously, it has been learned that the algae-derived phytonutrient astaxanthin—an outstanding scavenging antioxidant for biological membranes—can also function as a PPARα agonist, for which reason it can exert clinically useful hypolipidemic effects [107,108,109,110,111]. Astaxanthin may thus be useful in AMD both as an agonist for PPARα, and for protecting the mitochondrial electron transport chain from oxidative damage [112,113,114].

Overall, these considerations suggest that a nutraceutical regimens providing physiologically meaningful doses of several of these agents:ferulic acid, melatonin, glucosamine, berberine, lipoic acid and astaxanthin may have important potential for prevention and control of the dry form of AMD. Table 1 depicts dosage schedules of these agents that have proved to be clinically active for certain applications in past research. The drugs metformin and fenofibrate could function as alternatives to berberine and astaxanthin in this regard—albeit astaxanthin has scavenging antioxidant activity not possessed by fenofibrate. Potentially, these agents could be studied in RPE cell cultures in which Alu RNA expression has been artificially boosted.

It might also be noted that the protective impact of macular pigmentation—comprised of the xanthophyll carotenoids lutein, zeaxanthin, and meso-zeaxanthin, available as nutraceuticals—on AMD risk, suggests that photo-oxidative damage to retinal photoreceptors either promotes RPE deficit of Dicer1, or amplifies the pro-inflammatory signaling downstream from Dicer1 deficiency [117]. The key question remains: why is Dicer1 selectively depressed in RPE cells in AMD? Nutraceutical strategies for correcting the RPE deficit of DICER1 might cut to the root of AMD pathogenesis, but what these might be remains unclear.

Mention should be made of recent evidence that the adverse impact of Alu RNA accumulation on RPE survival requires cytoplasmic reverse transcription of this RNA into DNA within the cytoplasm [118]. This is mediated by the L1 reverse transcriptase, which can be inhibited by clinical concentrations of nucleotide reverse transcriptase inhibitor drugs used in management of HIV. An epidemiological analysis suggests that patients using such drugs regularly may be at lower risk for AMD [118]. Hence, a pharmaceutical strategy for managing dry AMD, potentially complementary to the measures suggested here, can be envisioned. However, rather perversely, these drugs have the potential to damage mitochondria [119].

## 4. Pertinence to Neovascular AMD

The neovascular, “wet” form of AMD is distinctly different from the dry form considered here. However, there is recent evidence that deficient genetic expression of Dicer1, whether general or specific to RPE cells, leads not only to geographic atrophy, but also to pathological neovascularization. Moreover, concurrent deficiency of proteins required for inflammasome signaling, or MyD88, required for IL-18 signaling, blunts this neovascularization [120]. This suggests that measures which oppose inflammasome activation in RPE cells—such as those suggested here—might also be expected to aid prevention of neovascular AMD.

Moreover, in light of a central role for endothelial NADPH oxidase activation in VEGF-mediated signaling that evoke retinal neoangiogenesis, the spirulina chromophore phycocyanobilin, a biliverdin derivative that can mimic the NADPH oxidase-inhibitory effects of its chemical relative bilirubin, may have potential for prevention and control of wet AMD [121,122,123,124,125,126]. Also, high intakes of glycine—which is clinically feasible, as this amino acid is highly soluble, pleasantly sweet, and inexpensive—has been demonstrated to have anti-angiogenic effects in rodent models of cancer and wound healing [127,128,129,130]. It has been suggested that indirect inhibition of endothelial NADPH oxidase activity may mediate this effect [131].

## Figures and Tables

**Figure 1 nutrients-14-01985-f001:**
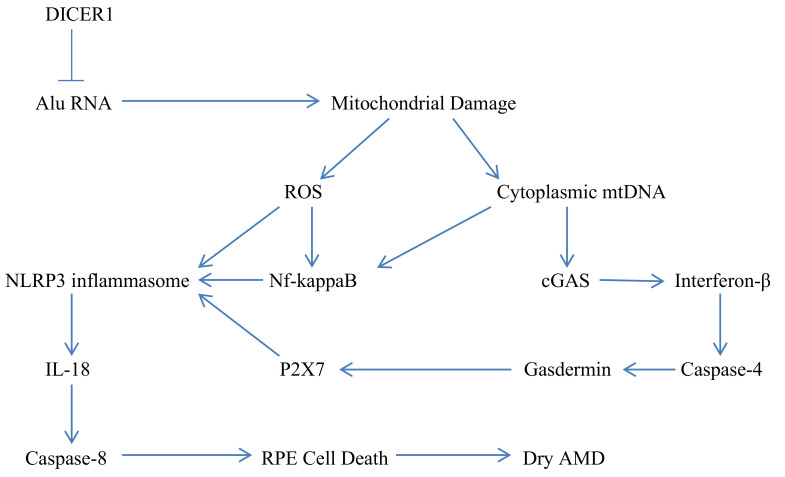
A simplified depiction of how DICER1 deficiency in RPE cells leads to Alu RNA excess, mitochondrial damage, NLRP3 inflammasome activation, autocrine IL-18 activity, and caspase-8-mediated cell death.

**Figure 2 nutrients-14-01985-f002:**
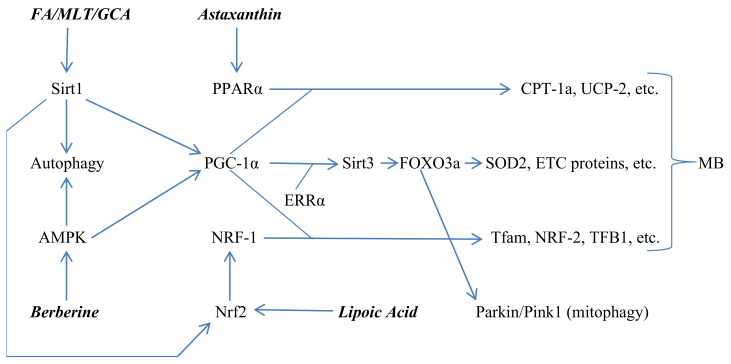
Roles for Sirt1, AMPK, Nrf2, and PPARα in promotion of mitochondrial biogenesis (MB) and autophagy/mitophagy. FA = ferulic acid; MLT = melatonin; GCA = glucosamine.

**Table 1 nutrients-14-01985-t001:** Suggested dose schedules for nutraceuticals with potential for controlling dry AMD.

Nutraceuticals	Dose Schedules
Ferulic Acid (or Sodium Ferulate)	250–500 mg twice daily [63]
Melatonin	3–20 mg at bedtime [115]
Glucosamine	1500–3000 mg once daily [81]
Berberine	500 mg, 2–3 times daily [99]
Lipoic Acid	600 mg 2 times daily [116]
Astaxanthin	12–20 mg daily [116]

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
