# Peer review of "Nutraceuticals/Drugs Promoting Mitophagy and Mitochondrial Biogenesis May Combat the Mitochondrial Dysfunction Driving Progression of Dry Age-Related Macular Degeneration"

_nutrients, 2022, doi:10.3390/nu14091985_

Round 1

Reviewer 1 Report

The paper by Dr. Lewis Lujàn is timely and nicely written. However, it appears more as a perspective rather than a review article. In the opinion of the present reviewer, the manuscript must be implemented in several parts. 

1) The authors must at least mention the pathogenic mechanisms of AMD with a special focus on the crucial role of RPE in the disease

2) Most importantly, the central part of the paper, which deals with the role of mitophagy/mitochondrial biogenesis, contains rather generalized information and no specific references supporting the beneficial role of mitophagy/mitochondrial biogenesis induction, included that by nutraceuticals discussed by the authors, in experimental/clinical AMD. The authors should implement this part by discussing studies carried out specifically in the frame of retinal diseases. The authors may consult the paper by Pinelli et al., 2020 IJMS, doi:10.3390/ijms21155563 as support .

3) At least a figure depicting the mechanisms which link DICER with mitochondrial alterations and nutraceutical/drug-targetable mechanisms within RPE-AMD appears mandatory to sum up the core message of the paper.   

Author Response

The paper by Dr. Lewis Luján is timely and nicely written. However, it appears more as a perspective rather than a review article. In the opinion of the present reviewer, the manuscript must be implemented in several parts. 

  • The authors must at least mention the pathogenic mechanisms of AMD with a special focus on the crucial role of RPE in the disease

We have added a new first paragraph in which we discuss the central role of RPE death in the pathogenesis of dry AMD.

  • Most importantly, the central part of the paper, which deals with the role of mitophagy/mitochondrial biogenesis, contains rather generalized information and no specific references supporting the beneficial role of mitophagy/mitochondrial biogenesis induction, included that by nutraceuticals discussed by the authors, in experimental/clinical AMD. The authors should implement this part by discussing studies carried out specifically in the frame of retinal diseases. The authors may consult the paper by Pinelli et al., 2020 IJMS, doi:10.3390/ijms21155563 as support .

We have revised this central part to cite evidence that anthocyanins (precursors to ferulic acid) and melatonin may reduce AMD risk.  Also, having recently encountered evidence that glucosamine is a Sirt1 activator, we have added a new paragraph discussing this, and also citing epidemiology correlating glucosamine use with reduced risk for AMD.  The Pinelli paper is now cited.

  • At least a figure depicting the mechanisms which link DICER with mitochondrial alterations and nutraceutical/drug-targetable mechanisms within RPE-AMD appears mandatory to sum up the core message of the paper.

This is a very good suggestion, given that the mechanisms whereby Alu RNA excess drive inflammasome activation are so complex.  So we have added a new Figure 1 which tries to clarify this graphically.

Many thanks for your helpful suggestions.

Reviewer 2 Report

In this manuscript the authors aim to review the literature discussing the beneficial role of some nutraceuticals/drugs in the activation of Sirt1, AMPK, Nrf2 and PPARa that can be exploited to induce the replacement of damaged mitochondria in dry age-related macular degeneration (AMD). The scope of this review is interesting, especially since there is not an efficient therapy for these patients. However, I found it too narrow. For example, the authors cite Dicer and Alu RNA, both involved in epigenetic mechanisms, one as a regulator the other as a target, but they do not explore the possibility of using nutraceuticals to control the epigenetics behind AMD.

Overall, the manuscript is well conceived but it is not really clear why the authors want to stress Alu RNA and Dicer involvement in mitochondrial damage if they are not the main targets of the nutraceuticals they want to review. Mitochondrial damage could be related more generally to the reduction of mitochondrial turnover during ageing. Moreover, references are often missing and at least one is incorrect. Below some other specific concerns that the authors should take into account to improve their manuscript:

-In the abstract the authors refer to age-related macular degeneration as “AMG”. It should be AMD as in the rest of the manuscript.

-Lines 43-44: Reference n.7 is incorrect. Ren et al. do not show accumulation of Alu RNA retrotransposon in RPE cells.

-Lines 44-48: References here are missing. Tarallo et al. Cell 2012 should be cited.

-Lines 56-58: Reference missing.

-Table 1: references to the indicated dosage of each nutraceutical should be added.

Round 2

Reviewer 1 Report

The authors adequately addressed the suggestions of the present reviewer. I have no further comments.

Reviewer 2 Report

The authors have addressed all my concerns.